# Evaluation of the Potential Hypoglycaemic Properties of *Mimusops zeyheri* Sond. and *Aloe marlothii* A.Berger, Two Plants Used by Traditional Healers in South Africa

**DOI:** 10.3390/plants13233323

**Published:** 2024-11-27

**Authors:** Ferdinand De Yogam Kamga-Simo, Guy Paulin Kamatou, Ananias Hodi Kgopa, Matlou Phineas Mokgotho, Leshweni Jeremia Shai

**Affiliations:** 1Department of Biomedical Sciences, Tshwane University of Technology, Private Bag X680, Pretoria 0001, South Africa; ferdinand.kamga@gmail.com (F.D.Y.K.-S.III); mokgothomp@tut.ac.za (M.P.M.); 2Department of Pharmaceutical Sciences, Tshwane University of Technology, Private Bag X680, Pretoria 0001, South Africa; kamatougp@tut.ac.za; 3Department of Biochemistry, Sefako Makgatho Health Sciences University, P.O. Box 236, Medunsa, Pretoria 0204, South Africa; ananias.kgopa@smu.ac.za

**Keywords:** glucose uptake, *Mimusops zeyheri*, *Aloe marlothii*, cytotoxicity, translocation, insulin signalling

## Abstract

*Mimusops zeyheri* Sond. And *Aloe marlothii* A.Berger are used traditionally in South Africa to manage many diseases, including diabetes mellitus. The mechanism through which these extracts exert blood glucose lowering is not well understood or reported. This study was aimed at assessing *M. zeyheri* and *A. marlothii* plant extracts for their potential to exhibit antidiabetic activity and their associated mechanisms. We evaluated the action of both extracts on major genes involved in the insulin signalling pathways in skeletal muscle cells. The in vitro cytotoxic effects of *M. zeyheri* and *A. marlothii* extracts were evaluated using the MTT assay and glucose uptake was evaluated using a glucose oxidase assay. The amount of translocated GLUT-4 was determined using the flow cytometry. Conventional PCR was used to determine the expression of GLUT-1 and GLUT-4 and RT-qPCR. IRS-1 total protein and Phospho-Akt were determined using ELISA. Plant extracts stimulated glucose absorption by skeletal muscle cells. *M. zeyheri* extract increased glucose absorption in muscle cells after 1 and 3 h of incubation. A 2-fold increase in translocated GLUT-4 was noted with *M. zeyheri*. The mRNA levels of GLUT-4 and GLUT-1 remained uniform in all treatments, while IRS-1, PI3K, Akt1, Akt2, and PPAR-γ were downregulated by both extracts. The expression of GLUT-4 was significantly increased by the action of insulin and *M. zeyheri* extract at 500 μg/mL. This study validates the traditional use of aqueous extracts of *A. marlothii* and *M. zeyheri* as hypoglycaemic plants and raises the assertion that the selected plant extracts utilise the IRS-1/PI3K/Akt pathway.

## 1. Introduction

Obesity, lifestyle, genetic predisposition and insulin resistance remain the main critical factors involved in the onset and progression of diabetes mellitus. Diabetes mellitus (DM) is a chronic disease which has become one of the biggest killer diseases of the 21st century [1]. Deutschländer et al. [2] define diabetes as a metabolic state characterised by hyperglycaemia and glucose intolerance, resulting from insulin deficiency, impaired efficacy of insulin action or both. Diabetes mellitus can be grouped into three main categories. Gestational diabetes mellitus is more common in pregnancy [3], while Type 1 diabetes mellitus, also called insulin-dependent diabetes mellitus (IDDM), results from the immunological destruction of the pancreatic β-cells, leading to insulin deficiency. Type 2 diabetes mellitus, also known as the non-insulin-dependent diabetes mellitus (NIDDM), is the most prevalent type and is associated with obesity, hereditary predisposition and is characterised by insulin resistance [4].

It has been proven that glucose absorption and utilisation in the muscle and adipose cells depends largely on glucose transporter 4 (GLUT-4) translocation from the cytoplasmic pool to the cell membrane [5,6]. This event also depends on the presence of stimuli such as insulin or exercise [7]. In normal cells, about 90% of GLUT-4 is located in the intracellular storage pools [8,9]. The uptake of glucose depends largely on two important signalling pathways, namely, AMP-activated kinase (AMPK) and phosphatidylinositol-3-kinase (PI3K) (AMPK/PI3K). These two physiological stimuli initiate signalling mechanisms that lead to the enhancement of GLUT-4 translocation and glucose uptake [10]. Insulin binds to its receptor, which phosphorylates insulin receptor substrates IRS1/2. These, in turn, activate PI3K, followed by the recruitment of serine–threonine kinase, Akt (protein kinase B), thus leading to the activation of downstream protein kinases. This activation results in GLUT-4 trafficking to the cell surface in order to facilitate glucose absorption [7,11]. Glucose transporter 1 (GLUT-1), on the other hand, is a facilitative glucose transporter protein expressed and active In most tissues, and accounts for basal glucose transport. Translocation of this protein is also activated by the presence of insulin [12].

Medications currently available for treatment of diabetes mellitus include insulin and numerous antidiabetic molecules, such as thiazolidinediones, sulfonylureas, metformin and glinides. Many of these medications are linked to serious side effects, making the search for new hypoglycaemic agents one of the key areas of exploration [13]. Furthermore, the toxicity associated with synthetic drugs has prompted consumers to use medicinal plants as alternatives. Plants present a rich source of secondary metabolites that may possess antidiabetic properties [14]. 

Evidence suggests that the use of plants to treat a wide range of diseases dates back thousands of years ago [15]. In South Africa, about 60–80% of the population uses medicinal plants for primary health care [16,17], partly due to their accessibility and affordability. However, the effectiveness, efficacy and toxicity of the plants generally used lack scientific evidence. 

*Mimosups zeyheri* Sond. (Sapotaceae), also known as “red milkwood”, is widely distributed from Tanzania in the north through to Zimbabwe, Angola, and Botswana to South Africa. In South Africa, the tree is found in the wild in Limpopo, Gauteng, Mpumalanga, North-West, Free State and KwaZulu-Natal provinces [18]. The roots of *M. zeyheri* are used in traditional medicine to treat ulcers and wounds [19], while the leaves are used for the management and treatment of diabetes mellitus [20]. This plant is also known to serve in nutritional developments in arid and semi-arid regions [18] and can contribute as a potential cure in the treatment of scurvy, particularly prevalent in the Limpopo region [21]. 

*Aloe marlothii* A.Berger (Asphodelaceae) grows mainly in bushveld vegetation, mountainous areas, rocky terrains and hills where temperatures are warmer and frost infrequent. It is widely distributed over Africa with hotspots of high diversity, occurring in Madagascar, East and West Africa, and especially southern Africa [22]. Its distribution in South Africa includes, the North-West, Gauteng, Limpopo, Mpumulanga and KwaZulu-Natal provinces [19]. The leaves of *A. marlothii* are used in South Africa to treat diabetes [20], while in the farming sector, it is used as a remedy to treat and prevent gallsickness, diarrhoea, constipation, helminthiasis, dystocia and to reduce tick burden [23]. Decoction of leaves and root is used in some parts of South Africa to treat African horse sickness and roundworm infestation, respectively [24]. The gel of this plant is used industrially in the fabrication of cosmetics, such as shaving and skin care creams, shampoos and for the treatment of skin diseases, particularly as a medicine for the treatment of burns [25]. 

In the present study, different concentrations of *M. zeyheri* and *A. marlothii* water extracts were prepared and their effects on glucose uptake and translocation of GLUT-4 assessed. Furthermore, the expression levels of *GLUT-4*, *GLUT-1*, Glyceraldehyde 3-phosphate dehydrogenase (*GAPDH*), insulin receptor substrate-1 (*IRS-1*), Phosphoinositide 3-kinase (*PI3K*), AKT Serine/Threonine Kinase 1 (*Akt1*), AKT serine/threonine kinase 2 (*Akt2*) and peroxisome proliferator-activated receptor gamma (*PPAR-γ*) were investigated in mouse C2C12 skeletal muscle cells. To the best of our knowledge, this study is the first to explore and report the antidiabetic activities and mechanisms of *M. zeyheri* and *A. marlothii* water extracts. This study is likely to contribute valuable information on the exact mechanism in which these extracts lower the blood glucose level. This information is likely to enrich the application and provide evidence on the use of medicinal plants for the development of novel antidiabetic drugs. Details of the medicinal plant species and plant parts selected for the study are presented in Table 1. 

## 2. Results and Discussion

### 2.1. Toxicity

Toxicity studies were conducted to assess the viability of the muscle cells, following a 48-h exposure to the plant extracts, using the MTT assay. Hydrogen peroxide, at 10% *v/v* dilution, reduced cell viability by 68%. *A. marlothii* and *M. zeyheri* extracts (5 mg/mL) had negligible toxicity against the muscle cells, with cell viability of 95% and 97% compared to the untreated cells, respectively. Instead, the two extracts, at 1.25 mg/mL, appeared to have mitogenic effects on muscle cells, increasing formazan intensity, an indicator of cell viability, by up to 21% higher than the untreated control cells (Figure 1), and the effect was dose-dependent. The concentrations needed to kill 50% of the cells (LC_50_) were all >5000 µg/mL, indicating relatively low in vitro toxicity of the two plants. These findings correlate with a previous toxicity study of *Aloe vera* on both HeLa and HepG2 cells, which showed that the whole leaf material of the plant was relatively safe with very high LC values of 413.9 and 439.0 mg/mL respectively, subsequent to a 4 h treatment [26]. Moreover, a toxicological study on rats by Mwale and Masika [27] revealed that the extract of *A. ferox* led to no mortalities, or behavioural or physiological changes in the animals. The authors concluded that *A. ferox* was relatively safe if used carefully. Furthermore, another report by Andersen [28] indicates that *Aloe vera-*derived ingredients were not toxic to mice, rats and dogs, with an LD_50_ greater than 200 mg/kg, 50 mg/kg and 50 mg/mL, respectively when administered peritoneally and 80 mg/mL, 15 mg/mL and 10 mg/mL, respectively when administered intravenously. Results from another study indicated that the methanolic leaf extract of *Mimusops elengi*, tested by shrimp and lethality bioassay, was relatively safe (LC_50_ = 80 μg/mL and LC_90_ = 320 μg/mL) [29], although the methanolic stem-bark extract was more toxic (LC_50_ = 40 μg/mL) [30]. Gami and co-workers [31] demonstrated that the acetate extract of *M. elengi* bark, at 5000 mg/kg body weight, was not toxic to mice. Furthermore, an investigation into the toxicity potential of the methanolic fruit extract of *M. kummel* revealed relative safety of the extract with median lethal dose (LD_50_) value greater than 2000 mg/kg in mice [32]. The same outcome was reached by other researchers [33], who reported no mortality or change in behaviour of the mice after an administration of up to 2000 mg/kg of the methanolic leaf extract of *M. elengi*, although doses greater than 1500 mg/kg resulted in abundant watery stools. 

### 2.2. Glucose Uptake

Skeletal muscle is a tissue where most glucose is absorbed and disposed [34]. In this study, the effects of *A. marlothii* and *M. zeyheri* water extracts on glucose uptake in muscle cells (C2C12) was assessed. However, with the exception of L6 cells, previous studies have reported minimal insulin-dependent glucose uptake and GLUT-4 translocation in muscle cells [35]. As shown in Figure 2A, it was revealed that 1 h exposure of cells to the aqueous extract of *M. zeyheri* 500 μg/mL and 1 mg/mL *M. zeyheri* extracts demonstrated a significant increase in glucose absorption by 20% and 26% higher than untreated controls, respectively (*p* < 0.001). *A. marlothii* treatment led to a marginal increase (1% and 8%) higher than untreated control cells for 500 μg/mL and 1 mg/mL, respectively (*p* < 0.001). Unsurprisingly, insulin on the other hand, appeared to stimulate an increase in glucose consumption by the cells to up to 41% higher than untreated controls after 1-h incubation (*p* < 0.001). The glucose absorption assessment carried out in this study is supported by the study which investigated the effect of four plant extracts (*Euclea undulata, Schkuhria pinnata, Pteronia divaricata* and *Elaeodendron transvaalense)* used traditionally in South Africa to treat and manage diabetes on C2C12 cells [36]. We found that *M. zeheyri and A. marlothii* extracts had the capacity to stimulate an increase of 26% and 8% glucose uptake, at 1 mg/mL after 1-h incubation, respectively, comparable to the hypoglycaemic activity of 162.2% and 107.5% exhibited by *Euclea undulata* and *Schkuhria pinnata* extract at 1-h incubation, respectively. Other plant species led to minute effect on glucose uptake by cells at the same concentration after 1-h incubation. Moreover, another study [37] on Karanjin isolated from *Pongamia pinnata* extracts demonstrated a significant stimulation of glucose uptake in L6 cells in a concentration-dependent manner, reaching a 1.9-fold increase at 25 µM after 16-h incubation. Insulin (100 nM), on the other hand, added 20 min prior to the glucose uptake assay, resulted in a 2-fold increase in glucose uptake compared to the basal state in L6 cells. High activity observed in the treatment with insulin for 20 min was suggested to be due to the 100 nM used coupled with the type of muscle cells [37].

The trend remained similar even after 3 h of Incubation when compared with the 1 h treatment at all concentrations tested. *A. marlothii* maintained glucose uptake stimulation at a concentration of 1 mg/mL. At this concentration, *A. marlothii* significantly (*p* < 0.05) stimulated a 23% increase in glucose uptake after 1 h treatment when compared to the untreated control cells (*p* < 0.05). Both 500 μg/mL and 1 mg/mL *M. zeyheri* extracts significantly exhibited dose-dependent activities by 18% and 21% when compared with the untreated control, respectively (*p* < 0.05). The activity of both *A. marlothii* and *M. zeyheri* were enhanced as the concentration increased. Figure 2C shows that *M. zeyheri* extracts effectively stimulated glucose uptake by muscle cells after 6-h incubation, though to a lesser extent than observed after a 3-h incubation period. Increases in glucose uptake of 14% and 11% were recorded at 500 μg/mL and 1 mg/mL, respectively. *A. marlothii* significantly (*p* < 0.05) stimulated a glucose uptake increase of 22% at 1 mg/mL and 18% (% of untreated control cells) at 500 μg/mL after 6-h incubation. The best performing plant species at both concentrations for 6 h was *A. marlothii*.

Fallah and co-workers [38] found that *A. vera* leaf gel significantly lowered blood glucose levels of patients with advanced Type 2 diabetes mellitus. The results obtained indicate that *Aloe* leaf gel improves glycaemic control in patients afflicted with advanced Type 2 diabetes mellitus needing insulin. Furthermore, the extract of *A. vera* significantly reduced the blood glucose level in diabetic rats [39]. *Aloe* species are the most recognised plants used by traditional healers in the treatment and management of diabetes in South Africa [40]. Some studies conducted on *Aloe* species, such as *A. arborescens, A. greatheadii* and *A. vera,* have confirmed their hypoglycaemic activity in streptozotocin (STZ)-induced diabetic rats [39,41]. A study on the antidiabetic effects of *A. greatheadii* leaf gel and *A. ferox* extract in streptozotocin (STZ)-induced diabetes rat models reported a decrease in blood glucose level and an increase in insulin secretion when compared to diabetic control group. Although the effects of *A. greatheadii* and *A. ferox* leaf gel extracts were not statistically significant, the authors concluded that the minor changes observed may be clinically important [41]. These results are consistent with our findings, suggesting that *A. marlothii* possesses hypoglycaemic properties involving the enhancement of glucose absorption by cells. Taking into consideration the results obtained after the incubation of both plant extracts investigated in this study, we can conclude that *A. marlothii* and *M. zeyheri* are both promising plants that can be sources of medicines for management of Type 2 diabetes mellitus. Furthermore, this study proposes that the hypoglycaemic effects of *A. marlothii* in diabetic patients may involve the enhancement of glucose absorption by cells, such as skeletal muscle cells. In another study on a comparative evaluation into the hypoglycaemic effect of the *A. vera* leaf extracts and metformin in alloxan-induced diabetic rats revealed the reduction of blood glucose level, after treatment with 200 mg/kg.b.w and 400 mg/kg.bw. This finding was comparable to the action of metformin administered at a concentration of 50 mg/mL. The authors concluded that treatment of rats with *A. vera* leaf extract could reverse alloxan-induced hyperglycaemia in animals [39]. Although *M. zeyheri* is widely used in traditional medicine across South Africa, there is a lack of scientific record to support its role as an antidiabetic medicine. However, the hypoglycaemic activity of other *Mimusops* species has been scientifically proven. For instance, Zahid and colleagues [42] reported a significant (*p* < 0.001) decrease in blood glucose concentration in normoglycemic and alloxan-induced diabetic rats, when orally administered with the methanolic extracts of flower and leaves of *M. elengi* at a dose of 100 mg/kg of body weight. The hypoglycaemic effect of the extracts was compared to that of the standard drug tolbutamide at a concentration of 100 mg/kg body weight. The authors attributed the resulting hypoglycaemic activity of the extract to the D-mannitol, β-sitosterol and β-sitosterol-D-glycosides contained in the flowers of *M. elengis* and the sterols, reducing sugar and tannins contained in the leaves of *M. elengis*.

### 2.3. GLUT-4 Translocation

Increased glucose uptake in skeletal muscles is attributed to the increase in cell surface concentration of GLUT-4 [43]. Consequently, this study investigated whether the antidiabetic effect of the selected plant species was due to the translocation of GLUT-4. As expected, treatment with 100 nM of insulin led to a 2-fold increase in the translocation of GLUT-4, as measured using flow cytometry. Treatment of cells with 1 mg/mL of *M. zeyheri* and *A. marlothii* extracts led to 2-fold and 1.9-fold increase in GLUT-4 translocation, respectively, compared to the untreated controls (stained). Similarly, treatment of cells with extracts of *A. marlothii* and *M. zeyheri*, at 500 μg/mL led to a 1.96-fold increase of GLUT-4 translocation with respect to untreated cells (stained) (Figure 3). The present results led to the suggestion that glucose uptake primarily depends on GLUT-4 translocation, and that the hypoglycaemic activity of plant extracts and the amount of GLUT-4 translocated are connected [44]. This study is in line with work conducted elsewhere [45], which reported that 4-hydroxypipecoic acid (4-HPA), isolated from *Peganum harmala,* enhanced translocation of GLUT-4 in a skeletal muscle cell model, in a time- and dose-dependent manner. Furthermore, karanjin (10 µM) from *Pongamia pinnata* induced up to a 50% increase in GLUT-4 translocation in L6 myotubes [37]. Based on the observed glucose uptake results, this study suggests that an increase in glucose uptake may be attributed to an increase in translocation of GLUT-4 from the intracellular storage pool to the cell membranes [43]. 

### 2.4. Gene Expressions

The effects of selected plant extracts on mRNA levels on the selected genes were investigated and the results are presented in Figure 4. The densitometric analysis of the agarose gel (Figure 4A), following RT-PCR amplification of GLUT-1 and GLUT-4 (Figure 4B), indicated the uniformity of GLUT-4 expression levels in all treatments except for the 500 μg/mL *M. zeyheri* extracts, which demonstrated a slight increase. These results suggest that the plant species under investigation stimulate glucose absorption by cells without affecting expression levels of GLUT-4. Furthermore, these results may suggest that plant extract-stimulated GLUT-4 translocation and subsequent glucose uptake require activation of existing GLUT-4 without the need for the synthesis of new GLUT-4. These findings correlate with another report [46], which demonstrated that thiazolidinediones do not increase the expression of GLUT-4 in rodent muscle or human muscles cells but enhance the expression of GLUT-1. The expression of GLUT-1, on the other hand, was altered by different treatments. For instance, insulin stimulated the upregulation of GLUT-1. The expression levels of GLUT-1 in untreated cells and cells treated with extracts of *M. zeyheri* and *A. marlothii* were comparable. These two plants stimulated an enhanced absorption of glucose, suggesting a steady supply of GLUT-1 as an important factor in the mechanism involved in the absorption of glucose. This is supported by the finding that a plant that downregulated GLUT-1 also failed to stimulate glucose absorption. Therefore, GLUT-4 alone cannot account for the increase in glucose uptake by cells treated with plant extracts. This correlates with the findings of Jaiswal and colleagues [37], which indicated that GLUT-4 protein levels remained uniformly expressed in the presence of karanjin in L6 myotubes. In addition, these findings correlate with another report [47], which established that green tea polyphenol extract increased the expression of GLUT-1 in liver cells. Similarly, extracts of *Tinospora crispa,* a plant used as antidiabetic medication in Thai traditional medicine, increased GLUT-1 mRNA level in L6 muscle cells [48]. In another study, GLUT-4 mRNA levels were unaltered following treatment of C2C12 muscle cells with *Cassia abbreviata* bark extracts and insulin [49], although other reports indicate the ability of insulin and several plant extracts to upregulate GLUT-4 mRNA levels [47,50].

### 2.5. Real-Time Polymerase Chain Reaction

A quantitative real-time polymerase chain reaction (RT-qPCR) method was used to quantify mRNA levels of key genes involved in the insulin signalling pathway and glucose utilisation, and the results are presented in Figure 5. The results of this study show the potential of insulin to increase the expression of GLUT-4 mRNA significantly, while *A. marlothii* did not affect the expression of this gene (*p* < 0.05). The lower concentration (500 μg/mL) of *M. zeyheri* led to a 4-fold improvement in the expression of GLUT-4 in comparison to the untreated control. Furthermore, the results obtained show that the extracts of both *A. marlothii* and *M. zeyheri* did not have any effect on the expression of insulin receptor substrate 1 (*IRS-1*), phosphoinositide 3- kinase (*PI3K*)*,* protein kinase B (*Akt/PKB*) and peroxisome proliferator–activated receptor γ (*PPAR-γ*) at all concentrations used. Treatment of cells with extracts and insulin appeared to downregulate the expression of *Akt-1* and *Akt-2*, when compared to the expression in control untreated cells. However, insulin upregulated the expression of *PI3-K* and *IRS-1* genes. 

The expression patterns of the *PPAR-γ* gene were not affected by treatment of C2C12 cells with 500 μg/mL of both extracts for a 3-h incubation period. The data revealed that high concentrations of extracts (1 mg/mL) downregulated *PPAR-γ* expression in C2C12 cells. Although PPAR-γ is poorly expressed in muscle cells, its study is of great importance since skeletal muscles account vastly for the glucose metabolism [51]. PPAR*γ* activation has proved to affect the insulin signalling cascade via direct regulation of the expression and/or phosphorylation of specific signalling molecules. Treatment of Type 2 patients with both rosiglitazone or thiazolidinediones (TZD), used clinically as insulin sensitizers, activates PPAR-*γ,* which potentiates the effect of insulin and increases the tyrosine phosphorylation of the insulin receptor and IRS-1 as well as inducing the activation of Akt/PKB resulting in a decrease in insulin resistance and serum glucose concentration of patients suffering from Type 2 diabetes [51].

Some reports indicate that insulin-stimulated glucose uptake by muscle cells is reduced in diabetic patients and diabetic rats, despite normal Akt-1 activation [7,52]. However, the stimulation and expression of *Akt-2* was impaired in muscles of obese and diabetic patients. The impaired phosphorylation of Akt-2 on Ser474 was associated with decreased expression of Akt2 and GLUT-4 proteins [7]. This highlights the vital role of Akt-2 over Akt-1 for glucose uptake and disposal. The translocation of GLUT-4 in the muscle cells occurs following a series of cellular membrane events initiated by insulin receptor signals [53]. The binding of insulin to its receptor leads to the involvement and phosphorylation of the IRS adaptor protein, which binds and interacts with PI3K, which in turn catalyses the synthesis of PIP3. PIP3 acts as a docking site for PDK1 and Akt that stimulates small GTPases, AS160 and RAL-GAP complexes (RGC), which induce the translocation of GLUT-4 and subsequent enhancement of glucose uptake by target cells [54]. 

Our findings on *M. zeyheri,* correlate with the work carried out previously by Dhanya and colleagues [55], which showed that quercetin, a citrus flavonoid, increased *GLUT-4* mRNA levels in L6 myotubes 14-fold higher than the untreated control cells. The same study demonstrated an upregulation in *Akt*, *PI3K* and *IRS* mRNA levels, although the expression of *CaMKK, AMPK* and *MAPK*, key molecules involved in the AMPK pathway, were also increased. The authors suggested a crosstalk between the insulin pathway and the AMPK pathway to stimulate the upregulation of *GLUT-4*. Based on RT-qPCR data, it would thus appear that *M. zeyheri* stimulates glucose uptake using a mechanism(s) involving activation and translocation of the GLUT-4 protein as well as enhancing the expression of the *GLUT-4* gene. *A. marlothii,* on the other hand, did not affect the expression of the *GLUT-4* gene. It can be suggested that *A. marlothii* stimulates glucose absorption using a mechanism that involves activation and translocation of existing GLUT-4 proteins, rather than enhancing the synthesis of new GLUT-4 proteins. Unlike insulin, these plant species did not affect *IRS-1* and *PI-3K* expression at a gene level, suggesting that they are not insulin mimetic and may act through a mechanism that excludes the upstream signals in the insulin signalling pathway. The involvement of other glucose transporters, apart from GLUT-4 cannot be ruled out. Therefore, more studies on the expression of key genes at the protein level, as well as the proportion of phosphorylated Akt-1 and Akt-2 may shed more light on the mechanism involved in glucose absorption stimulated by plant extracts.

### 2.6. Protein Expression of IRS-1 and Akt

Insulin receptor subtrate-1 and Akt have been reported to play a central role in the insulin signalling pathways [55,56]. Phosphorylation of IRS and Shc proteins represent the point of attachment of several intracellular proteins containing Src-homology-2 (SH-2) domains [11]. Furthermore, activation of Akt stimulates the phosphorylation of Mtor, glycogen synthase-kinase-3β (GSK-3) and other downstream substrate resulting in GLUT-4 translocation, glucose absorption and a wide range of biological functions [57]. In addition, the inhibition of the activity of these important molecules may lead to reduced insulin sensitivity and insulin resistance in muscle cells [58]. To further assess the mechanism associated with the hypoglycaemic effect of *M. zeyheri* and *A. marlothii* extracts, we determined the protein levels of IRS-1 and Akt in C2C12 cells using an enzyme-linked immunosorbent assay (ELISA) kit, and the results are presented in Figure 6. *M. zeyheri* extracts significantly increased the expression of IRS-1 protein compared to the untreated control cells (*p* < 0.05). *A. marlothii* also stimulated the expression of IRS-1, although the effect was not significant. Insulin, on the other hand, significantly increased IRS-1 protein expression (*p* < 0.05). Treatment of the cells with the selected extracts and insulin resulted in increased phosphorylation of the Akt protein, although the effect was not significant. 

Our findings corroborate with prior work on mulberry leaf tea (*Folium mori*), a medicinal plant used to manage diabetes in Asian nations, which exhibited similar results by increasing IRS-1, PI3K, Akt and GLUT-4 protein level in 3T3-L1 adipocytes [59]. Previous work has reported impaired glucose metabolism among db/db mice due to a low level of IRS-1 phosphorylation [60]. Administration of *Ishige okamurae* extract, a brown alga, reduced blood glucose in db/db mice by increasing protein expression of IRS, PI3K and Akt in skeletal muscles and white adipocyte tissue of diabetic mice [61]. Similarly, flavonolignans, extracted from *Silybum marianum* seeds, suppressed palmitate-induced decrease of insulin-stimulated Akt Ser473 phosphorylation on palmitate-induced insulin resistant C2C12 myotubes [62]. Consequently, the action of the selected plant extracts raises the assumption that they may exert their hypoglycaemic actions through the activation of the IRS-1/PI3/Akt pathways.

The chemical composition of *Mimusops zeyheri*, especially the water extracts, is yet to receive noteworthy attention. However, phenolic compounds, such as Gallic acid, Catechin and Tannic acid, have been identified in hexane extracts of *M. zeyheri* [63]. Gallic acid has been found to improve hypoglycaemic status in a rat model, while also improving dyslipidaemia caused by a high fat diet [64]. It is not immediately clear if these compounds are the major hypoglycaemic agents in our study, especially in view of the fact that water, and hexane, was used as an extractant. The antidiabetic activity of Aloe species is well-known and may be attributed to various polyphenols present in the species. For instance, Aloesin and Aloesinol, when administered to diabetic rats, resulted in low circulating blood insulin and glucose, as well as increased Adiponectin concentration [65]. 

## 3. Conclusions and Recommendations

This study offers an interesting research insight that elucidate an extensive glucose utilisation mechanism expressed in muscle cells exposed to the extract of *M. zeyheri* and *A. marlothii*. Six important findings emerged from this research: Firstly, a slight toxicity was observed when muscle cells were treated with selected plant extracts at a concentration of 5 mg/mL. Secondly, plant water extracts stimulated glucose uptake at both concentrations tested, regardless of duration of incubation. Thirdly, both plants stimulated GLUT-4 translocation at all concentration tested. Fourthly, GLUT-1 contributes to the enhancement of glucose uptake through an increase in its expression. Fifthly, both plant extracts downregulated the expression IRS-1, PI3K, Akt/PKB and PPAR-γ genes and this implies that these extracts are not insulin mimetic and therefore regulate GLUT-4 translocation through an insulin-independent mechanism. Lastly, the increased protein levels of IRS-1 and enhanced phosphorylation of Akt raises the assertion that selected plant extracts utilise the IRS1-1/PI3K/Akt pathway in exerting their antidiabetic effects. The study of other molecules involved in insulin signalling will assist in determining the exact mechanism underlining the hypoglycaemic effect of *M. zeyheri* and *A. marlothii*.

## 4. Materials and Methods

### 4.1. Plant Material Collection and Identification

*Mimosups zeyheri* and *A. marlothii* leaves were collected at the South African National Biodiversity Institute (SANBI) in summer, 2017. Voucher specimens (TUT 132/2017 and TUT 133/2017) of these plants were deposited in the Department of Biomedical Sciences, Tshwane University of Technology, South Africa. The plants at the South African National Botanical Institute (SANBI) are identified by name tags attached to their stems and confirmed by Dr GP Kamatou, who is a botanist. 

### 4.2. Plant Extracts

The dried leaves were ground using a pestle and mortar, followed by grinding into a fine powder using a Polymix-Kinematica mill Lasec, Cape Town, South Africa), before extraction with distilled water. The powdered plant materials were soaked in distilled water for 24 h with constant shaking, at 100 rpm, on an orbital shaker (Lasec, Cape Town, South Africa). The extracts were filtered using Whatman No.1 filter paper and the filtrates were combined, frozen at −70 °C and freeze-dried for 3–5 days using a SP Scientific Virtis freeze dryer (United Scientific, Johannesburg, South Africa). The extracts were dissolved in 10% DMSO in phosphate buffered saline (PBS), pH 7.2, to a concentration of 10 mg/mL. 

### 4.3. Cell Culture

C2C12 cells were used for various assays such as glucose uptake, cytotoxicity, gene expression, as well as GLUT-4 translocation. The C2C12 cells (ATCC CRL-1772) were purchased by Dr A Kgopa from Separation Scientific, Johannesburg, South Africa. C2C12 cells were grown and maintained in RPMI 1640 (Highveld Biological, Johannesburg, South Africa), supplemented with 10% foetal bovine serum (FBS) and 5% penicillin–streptomycin antibiotic cocktail (*v*/*v*). Cells were cultured in T25 mL flasks, and the growth medium was changed three to four times a week. Cells were sub-cultured into a T75 mL flask once they reached 80% confluence. Incubation was carried out in a humidified atmosphere of air and 5% CO_2_ at 37 °C. The cells seeded into a 96-well plate were used for glucose uptake and cytotoxicity testing, while those seeded into 6-well plates were used for gene expression and flow cytometry assays. 

### 4.4. MTT Assay

The in vitro cytotoxic effects of the crude *M. zeyheri* and *A. marlothii* water extracts were evaluated on muscle cells using the 3-(4,5-dimethyl-2-Thiazyl)-2,5-diphenyl-2H-tetrazoluim bromide (MTT) assay [66]. About 1.0 × 10^5^ muscle cells/well were seeded into a 96-well plate and incubated for 72 h at 37 °C in a 5% CO_2_ humidified incubator. Thereafter, cells were treated with 100 µL of specific water extracts to achieve concentrations of 0.625, 1.25, 2.5 and 5 mg/mL. Hydrogen peroxide (10% *v*/*v*) was used as a positive control. The cells were then incubated for 48 h at 37 °C in a 5% CO_2_ atmosphere. Then, 10 μL MTT (5 mg/mL in phosphate-buffered saline, pH 7.2) was added to the wells and the plates incubated for a further 4 h at 37 °C for MTT reduction, followed by washing in PBS (pH 7.2). Thereafter, the formazan crystals were dissolved in DMSO, and absorbance was read at 570 nm using a microplate reader spectrophotometer (ThermoFisher Scientific, Johannesburg, South Africa). Untreated cells were used as a negative control. The percentage (%) cell viability was calculated using the equation below and expressed as % of negative controls:$$\%\;Cell\;viabilty=\left(\frac{A_{sample}}{A_{control}}\right)\times 100$$

### 4.5. Glucose Uptake Assay 

Glucose uptake was determined using a modified version of a previously described method [67]. Briefly, C2C12 cells, cultured as previously described, were dislodged through brief exposure to 0.25% trypsin in PBS. Cells were counted using a Bio-Rad TC-20 cell counter (Bio-Rad Laboratories, Johannesburg, South Africa) and resuspended in complete pre-warmed growth medium (RPMI-1640 containing 10% FBS, 4.5 g/L glucose and antibiotics). Cells, at a density of 2.5 × 10^3^/well were seeded in sterile 96-well plates and incubated at 37 °C for 4 days to differentiate into myotubes. Thereafter, 100 μL of plant extracts or insulin prepared in a complete medium, as described above, was added to the respective wells to yield a final concentration of 500 μg/mL and 1 mg/mL. The cells were then incubated for 1 h, 3 h and 6 h at 37 °C. Thereafter, 50 μL of the supernatant was transferred from each well to a clean 96-well plate and 100 μL of glucose oxidase reagent (Sigma-Aldrich [Merck Life Science], Johannesburg, South Africa) was added to each well. The plates were incubated at 37 °C for 15 min and the absorbance was read at 540 nm using a microplate spectrophotometer (ThermoFisher Scientific, Johannesburg, South Africa). Cells treated with 100 nM insulin were used as positive controls, while the untreated cells were used as negative controls. To calculate glucose uptake, the amount of glucose left in the medium after incubation was subtracted from that of the untreated cells and expressed as a percentage of the controls. The glucose uptake in the controls was considered 100%. The amount was expressed as a % of glucose in untreated cell wells.
$$\%\;Glucose\;uptake=\left(\frac{{A_{control}-A}_{sample}}{A_{control}}\right)\times 100$$

### 4.6. GLUT-4 Translocation Analysis

C2C12 skeletal muscle cells were grown in a 6-well plates to confluency before treatment with 500 μg/mL and 1 mg/mL of selected plant extracts and insulin solution (100 nM) and incubated for 3 h at 37 °C. Thereafter, cells were washed with ice-cold PBS, followed by 30 min fixing with 3% formaldehyde at room temperature. Then, cells were washed thrice in PBS, followed by blocking with 1:10 dilution of goat serum and 3% bovine serum albumin (BSA) (W/V) in PBS at 4 °C for 15 min in the dark. Subsequently, PBS was used to wash the cells three times. The cells were then incubated with 1:4000 dilution of rabbit anti-mouse GLUT-4 antibody-conjugated with FITC, in PBS, for 18 h at 4 °C in the dark. Thereafter, the cells were washed four times with ice-cold PBS and re-suspended in 300 µL of PBS. Following thorough washing in PBS, cells were wrapped in foil and stored in the fridge (4 °C) or analysed using a flow cytometer flow cytometer. 

### 4.7. Gene Expression

#### 4.7.1. RNA Extraction

Muscle cells, maintained as described previously, were cultured in a 6-well plate and incubated 37 °C for 3 h in the presence and absence of extracts of *M. zeyheri* and *A. marlothii*. Cells were then centrifuged at 2500 rpm for 5 min, and the pellet retained for total RNA extraction. Total RNA was extracted from the muscle cells using GeneJet RNA purification kit, according to the manufacturer’s (ThermoFisher, Johannesburg, South Africa) instructions and the purified RNA used immediately or stored at −70 °C until use.

#### 4.7.2. cDNA Synthesis

Complementary DNA (cDNA) was synthesised from each RNA sample (1 µg) using a RevertAid First Strand cDNA synthesis kit, following the recommendations of the manufacturer (ThermoFisher, Johannesburg, South Africa). The composition of the reaction mixture for preparation of cDNA included 10 μL total RNA used as a template and 2 μL of Oligo(dT) primer. To this mixture, 4 μL of 5 × reaction buffer, 1 μL of Riboblock RNase inhibitor (20 U/µL), 2 μL of 10 mM dNTP mix and 1 μL RevertAid M-MuLV reverse transcriptase (200 U/µL) were added. The reaction mixture was gently agitated, centrifuged for 5 s at 2500 rpm, and incubated for 1 h at 42 °C. The reaction mixture was heated to 70 °C for 5 min to inactivate reverse transcriptase.

#### 4.7.3. Conventional PCR Conditions

The cDNA derived from reverse transcription of mRNA was amplified in an Applied Biosystems thermal cycler (ThermoFisher, Johannesburg, South Africa). The composition of the master amplification mixtures included 25 μL Dream Taq Green PCR Master Mix (2×), 1 μL of each forward and reverse primer, 5 μL template DNA and 18 μL nuclease-free water. The amplification was initiated by denaturation of cDNA at 95 °C for 3 min, followed by 30 cycles of 95 °C for 30 s (denaturation), 30 s annealing at a specific annealing temperature for the primer set, and 72 °C for 1 min (extension), and a 7 min final extension step at 72 °C completed the PCR run. The annealing temperatures were 59 °C for *glut-1* and *GAPDH*, and 65 °C for *glut-4*. The PCR products were separated on 2% agarose gels run at 60 V for 30 min.

#### 4.7.4. Quantitative Real-Time Polymerase Chain Reaction (IRS-1, PI3K, Akt1, Akt2, PPAR-γ, and GLUT-4)

Complementary DNA derived from reverse transcription of mRNA was analysed using real time quantitative PCR (RT-qPCR), using RNA-direct SYBR Green real time PCR master mix (Accuris qMax Green qPCR Master Mix 2×). The primers sequences and target genes are shown in Table 2. The qPCR reaction was set up as follows: Template-cDNA (6.8 μL); 2× SYBR Green Mix (10 μL), Forward primers (0.8 μL), Reverse primers (0.8 μL), H_2_O (1.6 μL) to make a total volume of 20 μL. The amplification was initiated by denaturation of cDNA at 95 °C for 3 min followed by 30 cycles of 30 s at 95 °C, 30 s at 65 °C and 1 min at 72 °C. A 7 min extension step at 72 °C completed the PCR run. Relative quantification of gene expression was calculated using the 2^−ΔΔCt^ method.

### 4.8. Phosphorylation of Akt

Phospho-Akt levels were determined using an enzyme-linked immunosorbent assay (ELISA) kit, following the recommendations of the manufacturer (Elabscience, Houston, USA, sourced from Biocom Africa, Centurion, South Africa). Confluent cultures of C2C12 skeletal muscle cells grown in 6-well plates were treated with 1 mg/mL of specific extracts and insulin solution (100 nM) for 3 h. The cells were repeatedly washed with PBS and lysed by the addition of cell lysis buffer containing protease inhibitors (1:10 dilution). The lysates were shaken at 8 °C for 30 min and then centrifuged at 13,000 rpm for 10 min at 8 °C. The supernatant was transferred into a centrifuge tube and used immediately or stored at −70 °C. Then, 100 μL aliquots of each sample (supernatants) were added into appropriate wells, covered and incubated overnight at 4 °C with constant agitation. The plates were inverted to discard the solution, followed by washing (4×) with 1× wash solution (provided in the kit). Thereafter, 100 μL rabbit anti-phospho-Akt (Ser473) antibody was added to each well and the plate incubated for 1 h at room temperature with constant shaking, followed by washing in wash buffer. Then, 100 μL of 1× HRP-conjugated goat anti-rabbit IgG was incubated for 1 h at room temperature with constant shaking. After discarding the antibody solution and 4× washes in wash buffer, 100 μL of TMB One-Step substrate reagent was subsequently added to each well, followed by incubation for 30 min at room temperature with shaking in the dark. Finally, 50 μL of stop solution was added to each well and the absorbance at 450 nm was immediately read using a microplate reader. 

### 4.9. Determination of Insulin Receptor Substrate-1 Protein Levels

Insulin receptor substrate 1 (IRS-1) total protein was determined using an ELISA kit, following the recommendations of the manufacturer (Elabscience, Houston, USA, purchased from Biocom Africa, Centurion, South Africa). Confluent monolayer cultures of C2C12 in 6-well plates were treated with 1 mg/mL of selected plant extracts and insulin solution (100 nM) for 3 h. The cells were repeatedly washed with PBS and lysed using cell lysis buffer containing protease inhibitors (1:10 dilution). The lysates were shaken at 8 °C for 30 min, centrifuged at 13,000 rpm for 10 min at 8 °C, followed by transfer of the supernatants into clean centrifuge tubes. Then, 100 μL of the standard working solution was added to the first two columns of the plate. The samples (100 μL) were added to other wells of the plate, which was covered with a sealer and incubated for 90 min at 37 °C. The liquids were removed from each well through inversion of the plates, and 100 μL of biotinylated detection antibody working solution was added to appropriate wells. The plate was covered with the plate sealer and incubated for 1 h at 37 °C. The antibody solution was decanted, and wells washed with 350 μL of wash buffer. One hundred μL of HRP-conjugate working solution was then added to each well, the plate covered with the plate sealer, followed by a 30 min incubation at 37 °C. The working solution was decanted, and each washed with 350 μL of wash buffer. This last step was repeated five times, after which 90 μL of substrate reagent was added to each well and the plates covered with a new plate sealer. This step was followed by further incubation of the plate for 15 min at 37 °C in the dark. Fifty µL of stop solution was added to each well and the absorbance of each well was read using a microplate reader set to 450 nm. 

### 4.10. Statistical Analysis

Data obtained from the study was presented as the means ± SD of two independent triplicate experiments. The statistical significance of differences between data sets was determined using Student t-tests. The accepted level for the results was significance was *p* < 0.05 (*) and *p* < 0.001 (**).

## Figures and Tables

**Figure 1 plants-13-03323-f001:**
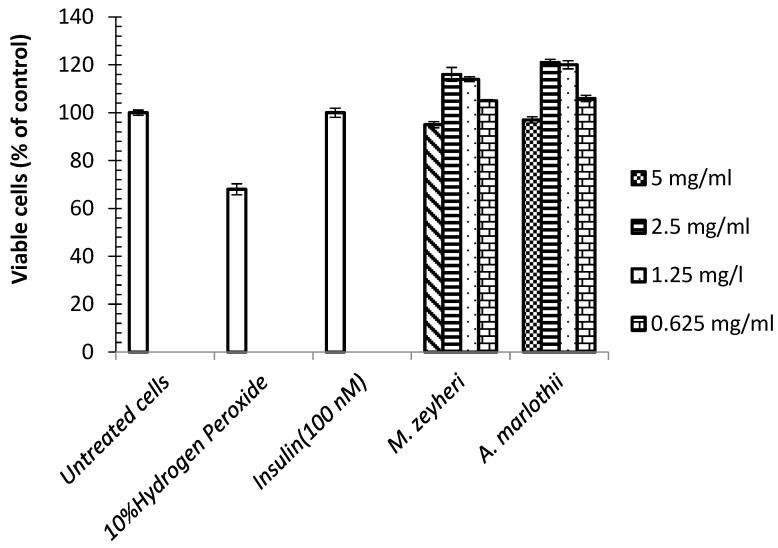
Toxic effects of *A. marlothi* A.Berger and *M. zeyheri* Sond. Water extracts on muscle cells after 48 h exposure. Data are mean ± SEM of three independent experiments performed in triplicates (*n* = 9). The results were expressed as percentage cell viability when compared to negative control cells.

**Figure 2 plants-13-03323-f002:**
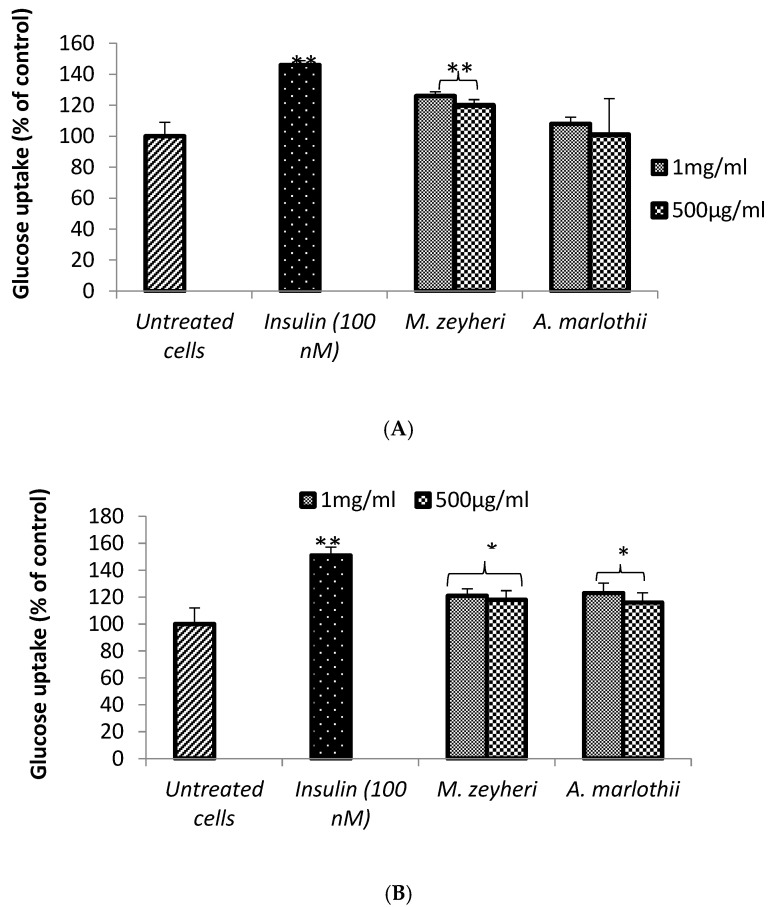
Hypoglycaemic activity of *A. marlothii* A.Berger and *M. zeyheri* Sond. Extracts on C2C12 cells treated with indicated concentrations for 1 h (**A**), 3 h (**B**) and 6 h (**C**). Data are means ± SEM of three independent triplicate experiments (*n* = 9). Results expressed as % glucose uptake. Statistically significant differences were considered when * indicates *p* < 0.05 and when ** indicates *p* < 0.001 when compared with the untreated control cells.

**Figure 3 plants-13-03323-f003:**
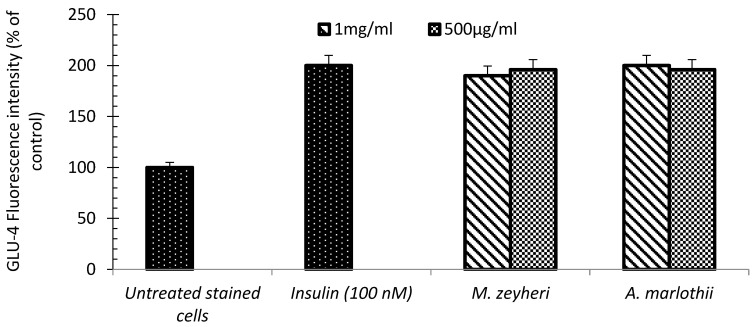
Quantitative analysis of GLUT-4 translocation in muscle cells after treatment with *A. marlothii* A.Berger and *M. zeyheri* Sond. Extracts. Data are mean ± SEM of three independent experiments performed in triplicates (*n* = 9). Results expressed as percentage fluorescence intensity. Insulin (100 nM) was used as positive control and untreated unstained cells as negative controls.

**Figure 4 plants-13-03323-f004:**
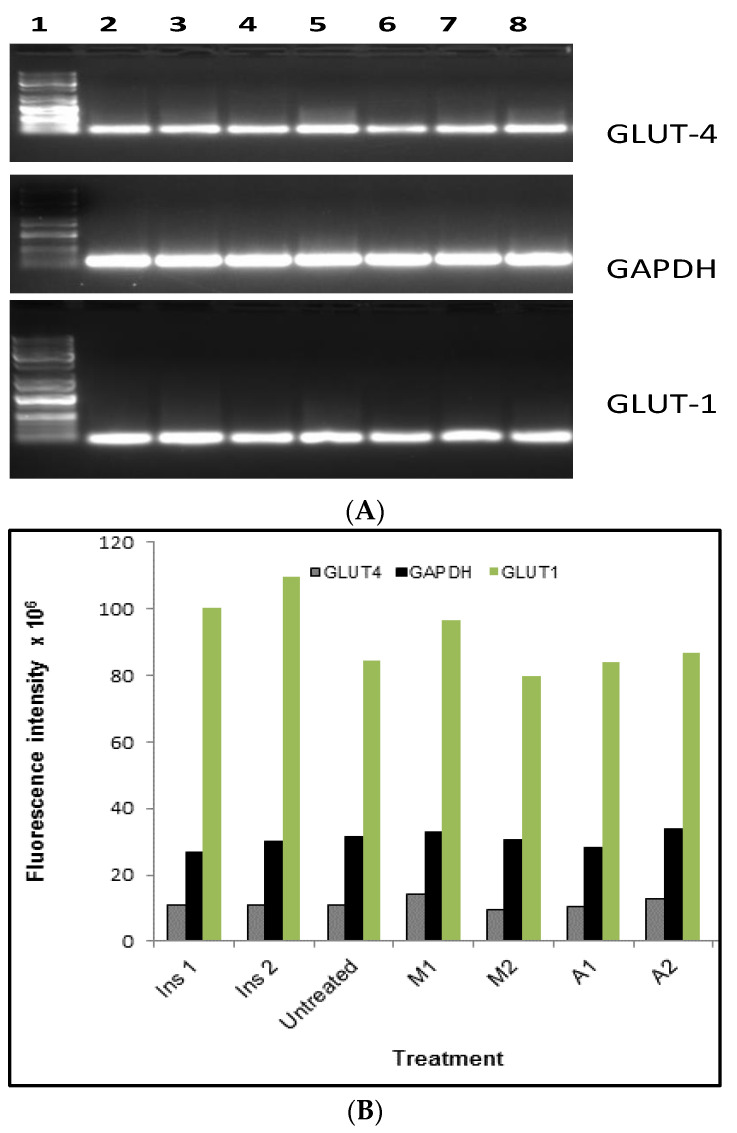
(**A**) RT-PCR product of *GLUT-1*, *GLUT-4* and *GAPDH* visualized in agarose gel electrophoresis and (**B**) densitometric quantification of the mRNA levels. Ins 1 and 2 = Insulin (100 nM and 50 nM), Untreated = Untreated cells, M1 and M2 = *Mimusops zeyheri* (500 μg/mL and 1 mg/mL), A1 and A2 = *Aloe marlothii* (500 μg/mL and 1 mg/mL). Agarose gel lanes: 1, MW markers, 2; 100 nM Insulin; 3, 50 nM insulin; 4, untreated cells; 5, 500 μg/mL *M. zeyheri*; 6; 1 mg/mL *M. zeyheri*; 7, 500 μg/mL *A. marlothii*; 8, 1 mg/mL *A. marlothii*.

**Figure 5 plants-13-03323-f005:**
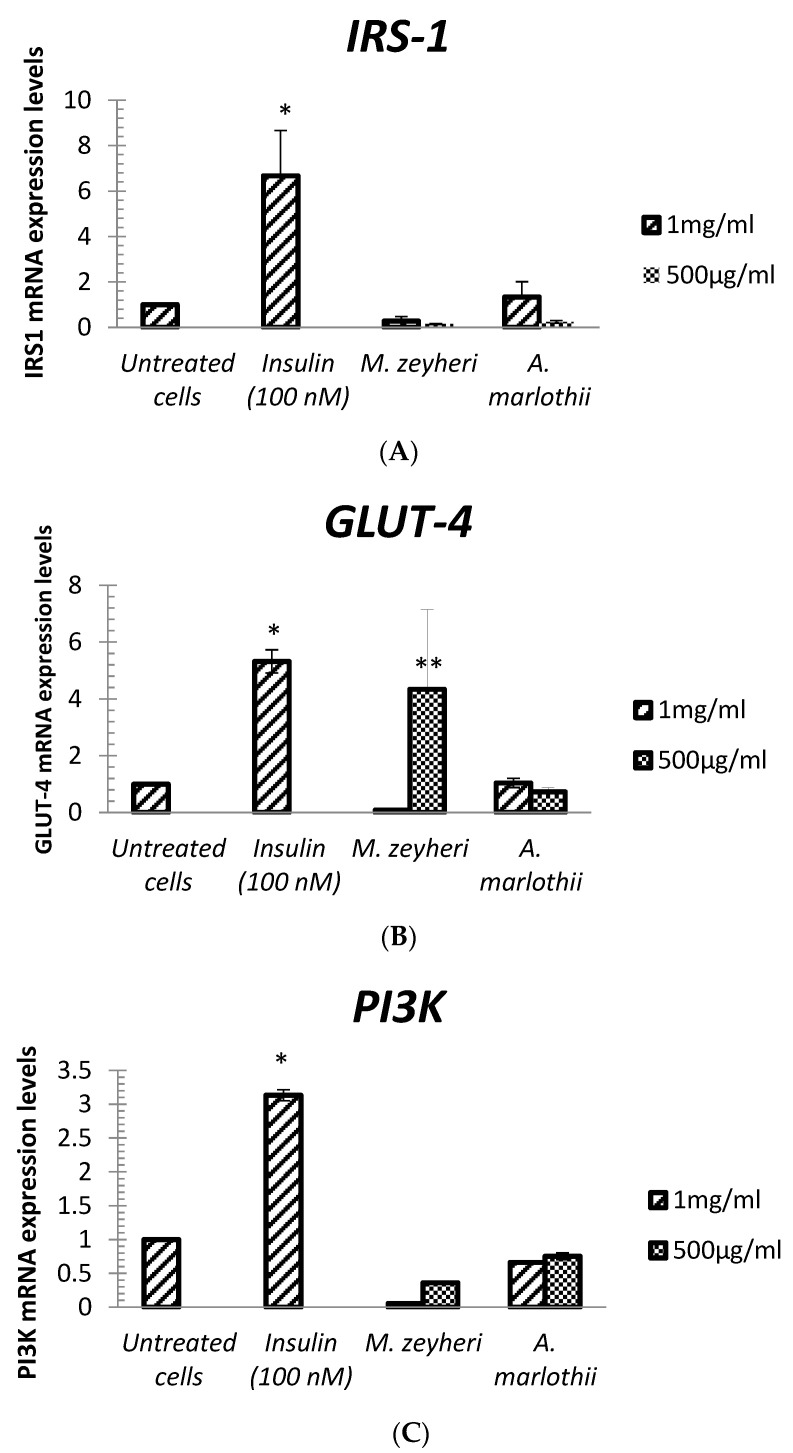
Expression levels (mRNA) of *IRS-1* (**A**), *GLUT-4* (**B**), *PI3K* (**C**), *Akt2* (**D**), Akt1(**E**), PPAR-γ (**F**) after treatment with extract of *A. marlothii* A.Berger and *M. zeyheri* Sond. Data are mean ± SEM of three independent triplicate experiments (*n* = 9). Results are presented as relative fold change in mRNA levels normalized to the *GAPDH*. Statistically significant differences were considered when * indicates *p* < 0.05 and when ** indicates *p* < 0.001 when compared with the untreated control cells.

**Figure 6 plants-13-03323-f006:**
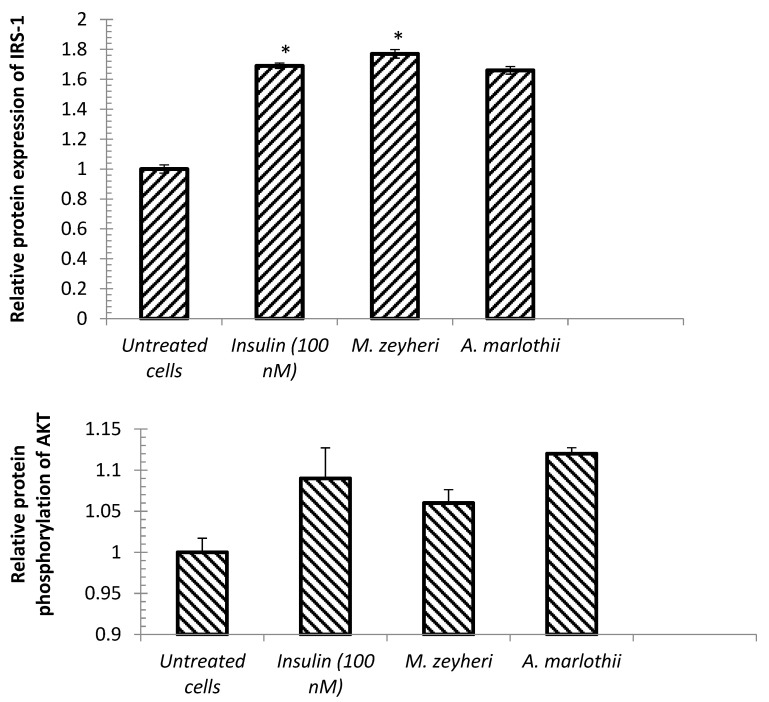
Effects of *M. zeyheri* Sond. and *A. marlothii* A.Berger extracts on IRS-1 protein expression levels and Akt phosphorylation in C2C12. Data are mean ± SEM of three independent experiments performed in triplicates (*n* = 9). Results are expressed as relative fold change in protein levels. Statistically significant differences were considered when * indicates *p* < 0.05 when compared with the untreated control cells.

**Table 1 plants-13-03323-t001:** Medicinal plants selected for the study [20].

Name of Plant	Family Name	Vernacular Name	Plant Part Used	Mode of Preparation
*Mimusops zeyheri* Sond.	Sapotaceae	Mmupudu	Leaf	Cooked for 10–25 min
*Aloe marlothii* A.Berger	Asphodelaceae	Kgopha	Leaves and roots	Cooked for 5 min

**Table 2 plants-13-03323-t002:** Genes analysed using RT-qPCR and their primer sequences.

Genes	Primers
*IRS-1*	Forward: TATCTGCATGGGTGGCAAGG Reverse: GGGTAGGCAGGCATCATCTC
*PI3K* *Akt1*	Forward: TGACGCTTTCAAACGCTATC Reverse: CAGAGAGTACTCTTGCATTC Forward: AAGGCCACAGGCCGCTACTA Reverse: AAGAAGAGCTCGCCCCGTT
*Akt2*	Forward: CGCCCTCTCGGTCGGTCTTCATCAG Reverse: TTCCAGCCATGAGCTACGTC
*PPAR-γ*	Forward: GGTGAAACTCTGGGATTC Reverse: CAACCATTGGGTCAGCTCTCTT
*GLUT-4*	Forward: CAACTGGACCTGTAACTTCATTGT Reverse: ACGGCAAATAGAAGGAAGACGTA
*GAPDH*	Forward: ACCACAGTCCATGCCATCAC Reverse: TCCACCACCCTGTTGCTGTA

Relative quantification of gene expression was calculated using the 2^−ΔΔCt^ method.

## Data Availability

Data are contained within the article. Further inquiries can be directed to the corresponding author.

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
