# Peer review of "Evaluation of the Potential Hypoglycaemic Properties of *Mimusops zeyheri* Sond. and *Aloe marlothii* A.Berger, Two Plants Used by Traditional Healers in South Africa"

_plants, 2024, doi:10.3390/plants13233323_

Round 1
Reviewer 1 Report
Comments and Suggestions for Authors
Remarks are provided as notes in the attached file.

Author Response
Comment 1: The table content does not match the table title.
Also, format the table so that all letters of the term are together. The Table has been corrected as suggested.
Comment 2: Format the entire manuscript according to the instructions for the author including units e.g. ml should be mL . The units have been changed as suggested. The Materials and Methods section was moved to correct part of the manuscript.
Reviewer 2 Report
Comments and Suggestions for Authors
-Table 1 should be include into the text.
-Lines 97-98 the abbreviation should be explain.
-Line 40 "Diabetes mellitus can be grouped into 3 main categories." Explain these 3 categories.
-There are data about antidiabetic actvity of studied plants? If yes, should be included into the text.
-Why did you analyse water extracts? Are you sure that this is the right solvents? Any references?
-Part 2.1. Who recognised these plants?
-Preparation od the plant extract is not clear. this part should be re-written.
-Some data should be in the tables and ANOVA should be performed.
-The conclusion should be re-written. Calculation should not be used.
Author Response
-Table 1 should be include into the text. Table 1 was already included in the text. The heading of the table was changed to be in line with suggestions by Reviewer 1.
-Lines 97-98 the abbreviation should be explain. The abbreviations were fully explained. However, this is generally not a standard practice.
-There are data about antidiabetic activity of studied plants? If yes, should be included into the text. We have not come across any report on the lab-based investigation of Mimusups zeyheri and Aloe marlothii as antidiabetics.
-Why did you analyse water extracts? Are you sure that this is the right solvents? Any references? Water was chosen as it is the solvent used by traditional healers for preparation of their concoctions. To validate the traditional uses, it was important to emulate the traditional methods, mainly on the use of the solvent.
-Part 2.1. Who recognised these plants? The plant materials were collected from a botanical garden, under the South African National Botanical Institute (SANBI). Each plant has a tag bearing its name, and they are all identified by the botanists at SANBI.
-Preparation od the plant extract is not clear. this part should be re-written. We have not changed this section, as we believe this is correct. It would have been easier if the changes to be made were indicated. As it is, we believe it reads as it should.
-Some data should be in the tables and ANOVA should be performed. We have performed student t-test, mainly to compare controls to the experimental treatments.
-The conclusion should be re-written. Calculation should not be used. The conclusion was modified. We are however not sure what reviewer meant by calculations.
Reviewer 3 Report
Comments and Suggestions for Authors
Dear editors, Dear authors, In the manuscript entitled: Evaluation of the potential hypoglycaemic properties of Mimusops zeyheri and Aloe marlothii, two plants used by traditional healers in South Africa (Ref. plants-3298712), the authors have applied proven methods to evaluate the antidiabetic effects of two traditionally used African plants that have been little studied previously. Most of the data are new and are likely to have some scientific significance. The results are adequately presented and well discussed. However, a revision of the manuscript is required. All suggestions and comments are included in the manuscript text in the appropriate places. I hope that these suggestions will improve the quality of the work.

Author Response
The review by reviewer 3 were important, as they contributed to the improvement of the manuscript.
We confirm that we have attended to all queries.
Figure 4 was corrected by inserting a new graph.
Figure 5 was formatted as suggested. The original submitted manuscript was inline with the reviewer's suggestions, and might have changed during formatting at the journal.
All suggestions were heeded.
Round 2
Reviewer 2 Report
Comments and Suggestions for Authors
In my opinion the manuscript in present form can be published in Plants.
Author Response
Thank you to the reviewer